# Simulation and Analysis of Perturbation and Observation-Based Self-Adaptable Step Size Maximum Power Point Tracking Strategy with Low Power Loss for Photovoltaics

**Yinxiao Zhu [1], Moon Keun Kim [2],\* and Huiqing Wen [1]**

[1] Department of Electrical and Electronic Engineering, Xi'an Jiaotong–Liverpool University, Suzhou 215123, China; Yinxiao.Zhu17@student.xjtlu.edu.cn (Y.Z.); Huiqing.Wen@xjtlu.edu.cn (H.W.)

[2] Department of Architecture, Xi'an Jiaotong–Liverpool University, Suzhou 215123, China

\* Correspondence: Moon.Kim@xjtlu.edu.cn or yan1492@gmail.com; Tel.: +86-512-8818-0465

**Abstract:** Photovoltaic (PV) techniques are widely used in daily life. In addition to the material characteristics and environmental conditions, maximum power point tracking (MPPT) techniques are an efficient means to maximize the output power and improve the utilization of solar power. However, the conventional fixed step size perturbation and observation (P&O) algorithm results in perturbations and power loss around the maximum power point in steady-state operation. To reduce the power loss in steady-state operation and improve the response speed of MPPT, this study proposes a self-adaptable step size P&O-based MPPT algorithm with infinitesimal perturbations. This algorithm combines four techniques to upgrade the response speed and reduce the power loss: (1) system operation state determination, (2) perturbation direction decision, (3) adaptable step size, and 4) natural oscillation control. The simulation results validate the proposed algorithm and illustrate its performances in operational procedures.

**Keywords:** perturbation and observation; adjustable step size; low power loss; maximum power point tracking

## 1. Introduction

### 1.1. Background

A direct current (DC) [1] pattern exists in almost all the electrical devices in our daily life. Photovoltaics (PV), as well-known renewable power generation solutions, are a foundational DC source which can supply DC power for DC application directly or drive the alternating current (AC) application after inverting. Owing to the policy support and sharp cost reduction of photovoltaic [2] techniques, solar power, a form of inexhaustible eco-friendly energy, has been widely exploited in daily life in recent years. At the same time, the civilization process enhances the demands of civil space. To increase the utilization of urban space, building-integrated photovoltaic techniques are becoming more widely considered in the research community [3–10].

Building integrated photovoltaics, a significant branch of PV generation, are easily affected by environmental conditions, similar to other PV applications. The output characteristics of the PV panel are mainly influenced by the illumination intensity, temperature, material, and other conditions, especially the received illumination intensity and the surface temperature of the PV panel. For example, increasing the temperature results in a slight increase in the short-circuit current and a significant decrease in the open-circuit voltage, which reduces the maximum output power.

However, in any condition, a curve of the output power can be drawn, and the output power has a maximum output point called the maximum power point (MPP) [7,11–22]. For the PV system operated with higher efficiency, an MPP tracking (MPPT) controller is indispensable; the tracking methodologies are introduced in detail in the MPPT section of this report.

For a PV system with an MPPT controller, the structure can be depicted as consisting of a PV panel, a power converter [7,8,16,19–33], an MPPT controller, and a load (including but not limited to motors, batteries, heaters, energy-storage systems, and other electric appliances). The structure of the PV system is shown in Figure 1.

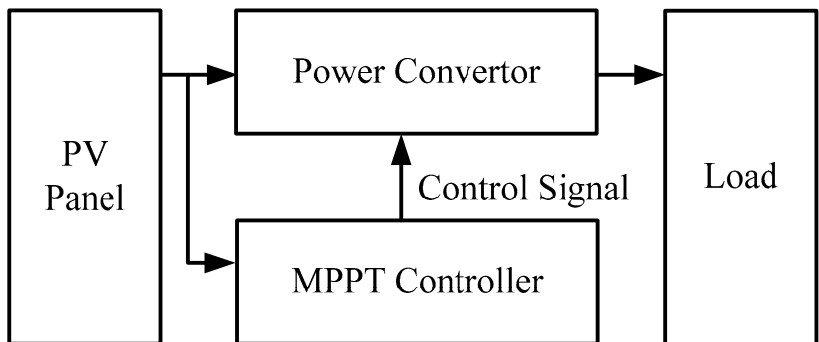

**Figure 1.** Structure of a photovoltaic (PV) system with a maximum power point tracking (MPPT) controller.

The operating process of the PV system can be simply explained as follows. After absorbing enough radiation, the PV panel supplies electricity to the power converter and the load is driven via the output from the power converter. Simultaneously, the MPPT controller measures specific parameters (such as voltage and current) for controlling the power converter in order to make the system operate at the MPP.

### 1.2. Aims and Objectives

This study discusses an advanced algorithm to improve the efficiency of the perturbation and observation (P&O)-based MPPT with simulation and numerical analysis tools. Power loss, a common phenomenon in electricity generation, refers to the power consumed during the conversion process; it is unavoidable, but can be reduced. For the conventional P&O-based MPPT controller of a PV system, certain power loss is caused by the ineluctable perturbation of the P&O method. If the power loss can be reduced, the utilization rate of the solar energy can be increased, and more energy can be saved.

To solve the power-loss problem caused by the non-environmental conditions causing oscillation, a P&O-based self-adaptable MPPT algorithm is designed in this study. This algorithm is expected to reduce the power loss and improve the response speed of tracking.

### 1.3. Model and Characteristics Analysis of PV Panel

The PV cell, also known as a solar cell, is the unit component of the PV panel and is a semiconductor device that can directly convert solar power into electrical energy based on the PV effect [34]. The irradiation directly affects the intensity of photocurrent generation, influencing the photovoltaics.

According to the one diode PV cell structure shown in Figure 2, a PV cell can be considered as the equivalent circuit depicted in Figure 2, consisting of an ideal current source ($I_L$) with a diode (D), a series resistor ($R_S$), and a parallel shunt resistor ($R_{Sh}$).

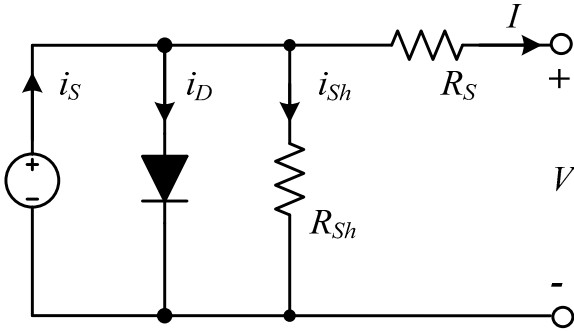

**Figure 2.** Equivalent circuit of a PV cell.

According to the equivalent circuit shown in Figure 2, the output current of a PV cell can be calculated as follows [6,26,35–39]:

$$I = I_L - I_D - I_{Sh} \tag{1}$$

Here, $I$ is the output current (A), $I_L$ is the photocurrent (A), $I_D$ is the diode current (A), and $I_{Sh}$ is the shunt current (A); $I_L$, $I_D$, and $I_{Sh}$ can be expressed as follows [6,26,35–39].

$$I_L = \mu G \tag{2}$$

$$I_D = I_0 \left\{ \exp\left[ \frac{V + IR_S}{n(kq/T)} \right] - 1 \right\} \tag{3}$$

$$I_{Sh} = \frac{V + IR_S}{R_{Sh}} \tag{4}$$

Here, $\mu$ is a proportional constant (depending on the material and other conditions), $G$ is the illumination intensity, $I_0$ is the diode reverse saturation current (unit: A), $n$ is the diode ideality factor ($1 < n < 2$, and $n = 1$ for an ideal diode), $k$ is Boltzmann's constant ($1.38 \times 10^{-23}$ J/K), $q$ is the elementary electric charge $e$ ($1.6 \times 10^{-19}$ C), $T$ is the absolute surface temperature of the PV cell (unit: K), $V$ is the output voltage, $I$ is the output current (unit: A), $R_S$ is the series resistor (unit: $\Omega$), and $R_{Sh}$ is the shunt resistor (unit: $\Omega$).

Combining Equations (2)–(4), as well as output current $I$, Equation (1) yields [6,26,35–39]:

$$I = \mu G - I_0 \left\{ \exp\left[ \frac{q(V + IR_S)}{nkT} \right] - 1 \right\} - \frac{V + IR_S}{R_{Sh}} \tag{5}$$

According to Equation (5), the temperature and illumination intensity are the most influential environmental conditions in actual operation, because the other uncertain factors are confirmed upon the completion of the PV cell.

The characteristic current to voltage (I–V) and power to voltage (P–V) curves under different temperature and illumination conditions are shown in Figure 3. Changes in the temperature and illumination can easily affect the MPP, but in different ways. As shown in Figure 3a, under the same illumination ($G$), the increasing temperature ($T$) visibly reduces the output voltage of the PV panel, but the decrease in the output current is limited. This is followed by a decrease in the output power. As shown in Figure 3b, at the same surface temperature, as the illumination increases, the output voltage exhibits a slight increase, but the output current increases sharply, followed by an increase in the output power.

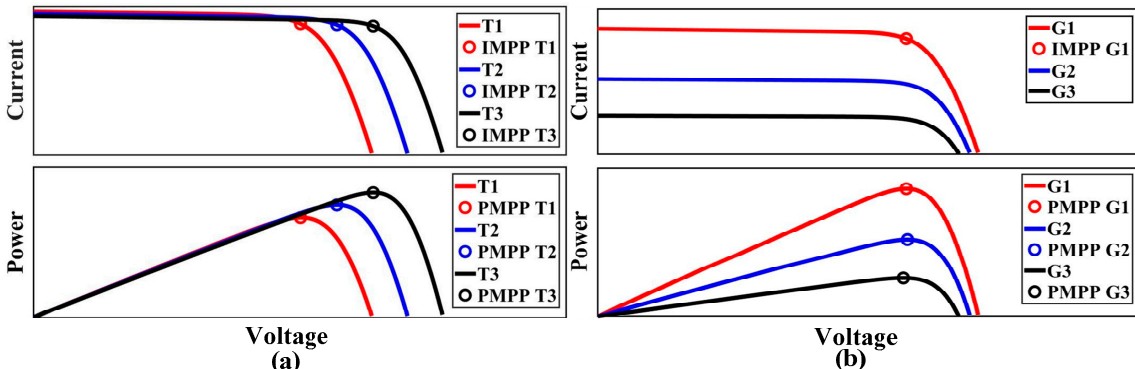

**Figure 3.** Current to voltage (I–V) and power to voltage (P–V) curves under variable conditions: (**a**) $T_1 > T_2 > T3$; (**b**) $G_1 > G_2 > G_3$.

### 1.4. Maximum Power Point Tracking (MPPT)

The MPPT technique aims to maintain the maximum output operation of the PV system. MPPs in different environmental conditions are marked in Figure 3.

In most cases, a pulse-width modulation (PWM) wave is the control signal for the power switch of the converter; the duty cycle (*D*) of the PWM wave affects the output voltage, and an MPPT controller controls the duty cycle of the PWM wave [40,41].

In actual operation, the environmental conditions do not change sharply every second; nonetheless, the MPPT controller is needed to achieve the MPP. The core of the MPPT controller is the MPPT algorithm. According to their characteristics, MPPT algorithms can be classified into self-optimization and non-self-optimization algorithms. For example, perturb and observe (P&O) [29,40,42], incremental conductance (InC) [25,43,44], and constant voltage tracking (CVT) [40,45] are three typical self-optimization algorithms. Non-self-optimization algorithms mainly include curve fitting [46] and other methods. Furthermore, there are artificial intelligence techniques, such as fuzzy logic [12,47,48] and particle swarm optimization [12,49–52], that are combined with conventional MPPT methods to achieve a high tracking accuracy.

In the industry, MPPT controllers, mostly self-optimization-based, can help systems track the MPP and automatically maintain steady operation in the maximum-output state. A comparison of three typical methods is shown in Table 1. After the comparison, to simplify the algorithm, the proposed MPPT algorithm is P&O-based [29,42,53], and its differences from the conventional one [13,16,30,36,45,46,53] are introduced in the Methods section of this report.

**Table 1.** Comparison of constant voltage tracking (CVT), incremental conductance (InC), perturb and observe (P&O), and the proposed maximum power point tracking (MPPT).

| MPPT Algorithm | CVT | InC | P&O | This Work |
|---|---|---|---|---|
| Specific PV Array | Yes | No | No | No |
| True MPPT | No | Yes | Yes | Yes |
| Tracking Speed | Adaptable | Medium | Adaptable | Fast |
| System Complexity | Low | Low | Medium | Medium |
| Measured Parameters | Voltage | Voltage, Current | Voltage, Current | Voltage, Current |

## 2. Methods

### 2.1. Principle of the P&O Method

The P&O method is the most widely used self-optimization MPPT algorithm. The basic principle of P&O is as follows. After a certain directional-changing voltage applies perturbation to the output voltage of the PV panel, the MPPT controller compares the output power before and after the perturbation. If the changing direction is positive and the output voltage increases, the MPPT controller

continues the perturbation in this direction; if the output power decreases, the direction reverses in the next perturbation.

Figure 4 shows the characteristic P–V curve, where $P_{MPP}$ is the MPP, $P_1$ is to the left of the MPP, $P_2$ is to the right of the MPP, and $\Delta U_1$ and $\Delta U_2$ are the changing ranges of the output voltage. To achieve the MPP, $\Delta U_1$ should be increased in $P_1$, but $\Delta U_2$ should be decreased in $P_2$. In this case, $\Delta U_1$ and $\Delta U_2$ differ, and $\Delta U_1 > \Delta U_2$. A greater distance from the MPP yields a greater difference between $\Delta U_1$ and $\Delta U_2$. Owing to the existence of perturbation, it is very difficult for the basic P&O method to eliminate the oscillating phenomenon at the MPP. The step size of the perturbation directly affects the speed and accuracy of the MPPT. All of these factors cause power loss. Figure 5 [40] presents the flowchart of the basic P&O tactic.

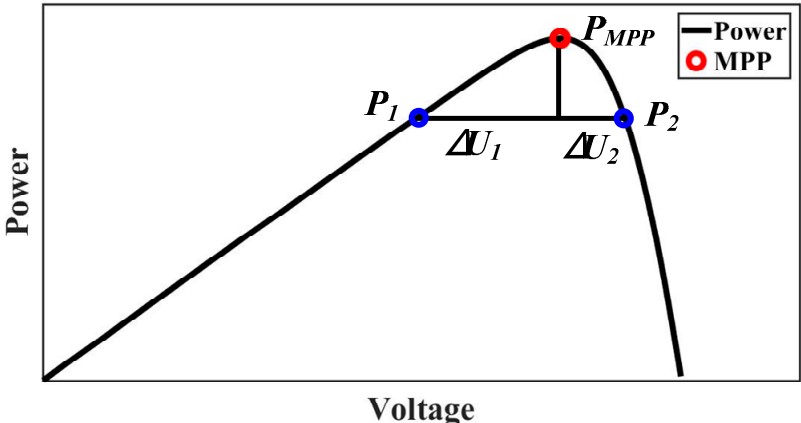

**Figure 4.** Tracking issues in the P–V curve.

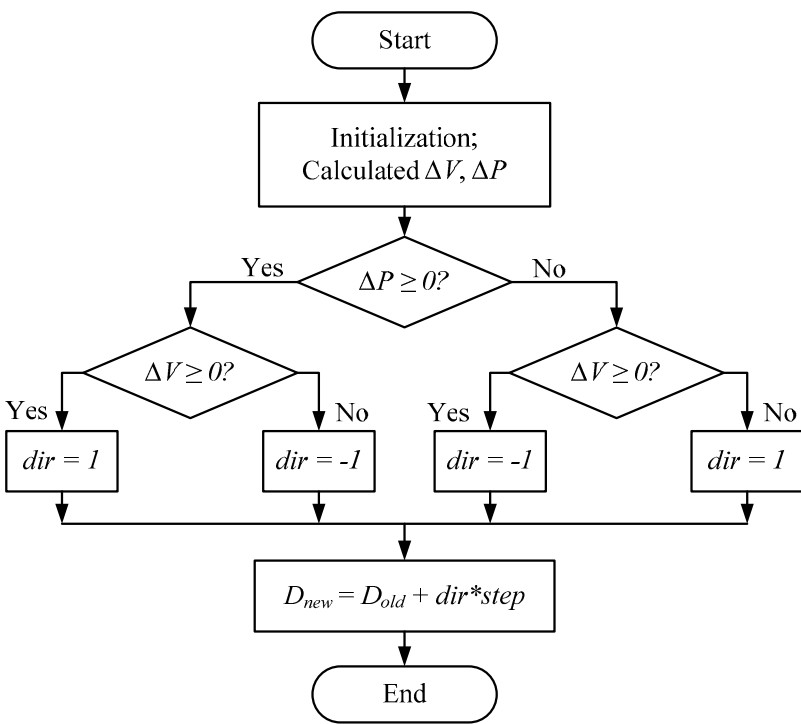

**Figure 5.** Flowchart of the basic perturb and observe (P&O) tactic.

## 2.2. P&O-Based Self-Adaptable Step Size MPPT Tactic

In the case of a fixed step size P&O algorithm, opportunely increasing the step size can improve the system response speed, but increase the oscillation region around the MPP and increase the power

loss. A small step size can increase the tracking accuracy and reduce the oscillation, but reduce the response speed. To deal with the contradiction between accuracy and response speed, the variable step size MPPT algorithm is employed. The conventional variable step size MPPT algorithm comprises the optimum gradient method [23,54], the successive approximation optimization method [29,30,42,43,55], and other methods. However, the derivative of the power to the voltage is too large on the right side of the MPP; therefore, the derivative of the power to the voltage is no longer suitable for the parameter of the step size solution. However, the optimal gradient-based variable step size MPPT uses a stationary step size selection equation; this algorithm cannot preferably adapt to changes in the P–V curve.

The tracking tactic of the conventional MPPT algorithm is periodic. For the conventional P&O strategy, the step size is fixed, which means that the $\Delta U$ in the operating procedure, shown in Figure 5, cannot change. Owing to the tracking issues presented in Figure 4, a certain oscillation exists. Because of the aforementioned issues, in the steady operation state, although the MPPT controller has tracked the MPP successfully, the output voltage still undergoes perturbation around the MPP and never achieves $V_{MPP}$ (output voltage in the MPP), as shown in Figure 6, and the exiting oscillation around the MPP causes certain power loss. The definition and analysis are introduced in the power-loss analysis and calculation part of this report. To deal with the power loss around the MPP in the steady-state operation as much as possible, an advanced P&O-based MPPT tactic with a self-adaptable step size is proposed. A flowchart of the proposed MPPT tactic is presented in Figure 7.

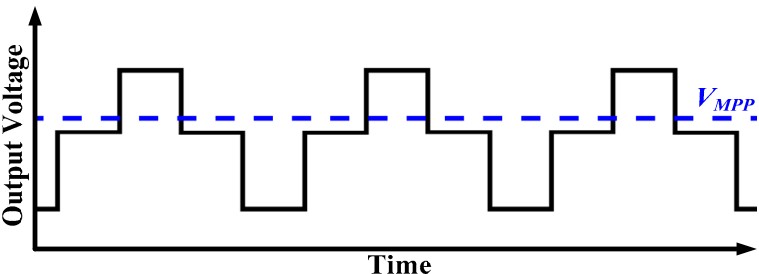

**Figure 6.** Oscillation in steady-state operation.

Compared with the conventional P&O MPPT [29,30,40,42], the improved procedure of the proposed P&O-based self-adaptable MPPT tactic is based on the following four key aspects: (1) natural oscillation control, (2) system operation state determination, (3) perturbation direction decision, and (4) adaptable step size.

In the natural oscillation control procedure, the proposed tactic can select a suitable tracking loop depending on the oscillation range. In the flowchart, $\Delta P$ and *err* detects the output power change in present and previous sample period; $E_{TH}$ is the threshold for error determination. It is used to control the allowable natural oscillation range and as an entry for a continuous module. If $err > E_{TH}$, the program uses the error value and multiplies it by a weight factor ($k$) as the step size to optimize the tracking speed; otherwise, the program enters the system operation state determination module.

In the system operation state determination module, *Flag* is the identifier of the operation state. If *Flag = 1*, the tracking procedure enters an idle operation loop, and the next perturbation director (*dir*) depends on whether the actual current ($I$) reaches the threshold for current ($I_{TH}$), expressed in Equation (7). If $\Delta I > I_{TH}$, *Idle* changes to 0, and the direction (*dir*) is a sine function of $\Delta I$ and is the weight of the next perturbation; or, it jumps out of the Idle Mode loop. If *Idle = 0*, the program enters the P&O-based perturbation direction decision procedure.

The procedure of perturbation direction decision is similar to the operation of the basic P&O tactic, but differences exist. The direction for the next perturbation depends on the change in the output power ($\Delta P$). If the output power in this perturbation is increased ($\Delta P > 0$), the perturbation direction is continuous, and the counter is initialized (*Cont = 0*); otherwise, the perturbation direction is changed (*dir = −dir*), and the loop time is counted (*Cont = Cont + 1*). After the conventional P&O procedure, the program determines the change in the output voltage ($\Delta U$) and the number of loop times (*Cont*).

If it loops more than once (*Cont > 1*) and the change in the output voltage is null (Δ*U = 0*), the program operates in the *Idle* mode (*Flag = 1*), provided that the error is within the allowable range.

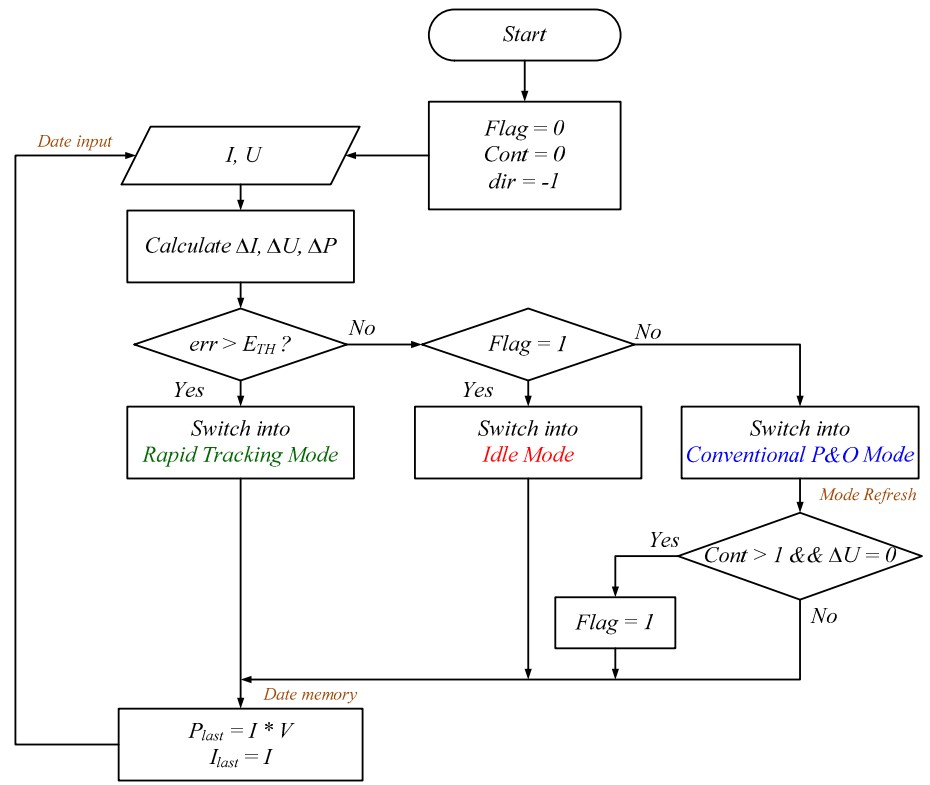

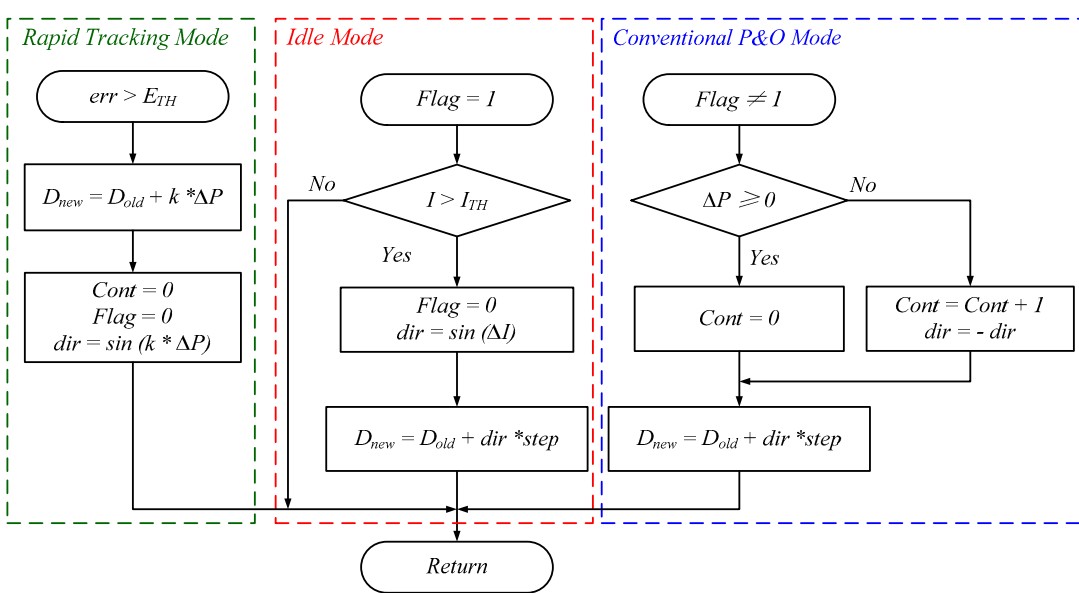

**Figure 7.** Flowchart of the proposed MPPT tactic.

As shown in Figure 6, the standard P&O tactic gives rise to a certain oscillation around the MPP, and the range of the oscillation depends on the setting of the perturbation step size. The adaptable step size is included in every aspect introduced above. Mainly, the step size depends on the change in the last operation state (*Flag = 0 or Flag = 1*) and the range of actual oscillation (*err*). The change in the step size affects the duty cycle of the PWM wave and is displayed as the change in the output

voltage of the boost converter. In this tactic, the step size is identified as *step* and can be measured by determining the output voltage or duty cycle of the PWM wave in actual operation.

The boundary condition chosen is based on the characteristics of selected PV elements as shown in Table 2. During the irradiation change, the change in interface temperature would not be significant. Hence, the consideration of boundary selection is only based on the change in MPP while the irradiation changes.

In the simulation, the boundary condition of $E_{TH}$ is selected by the change in measure power during the no-oscillation state. The value of $E_{TH}$ is selected as 0.03; in other words, the judgement follows the relationship of $|(P\text{-}P\_old)/P\_old| >= 0.03$ ($P$ is the present sampled power and $P\_old$ is the value in previous sample time). $|(P\text{-}P\_old)/P\_old|$ is the explaination for *err*. The selection of 0.03 is based on the change in power while the irradiation changes for the PV element MSX-60W, as shown in Figure 8. According to Figure 8, if the change in power is more than 0.03 of the previous MPP point, the change in irradiation could be determined and the Rapid Tracking Model is activated.

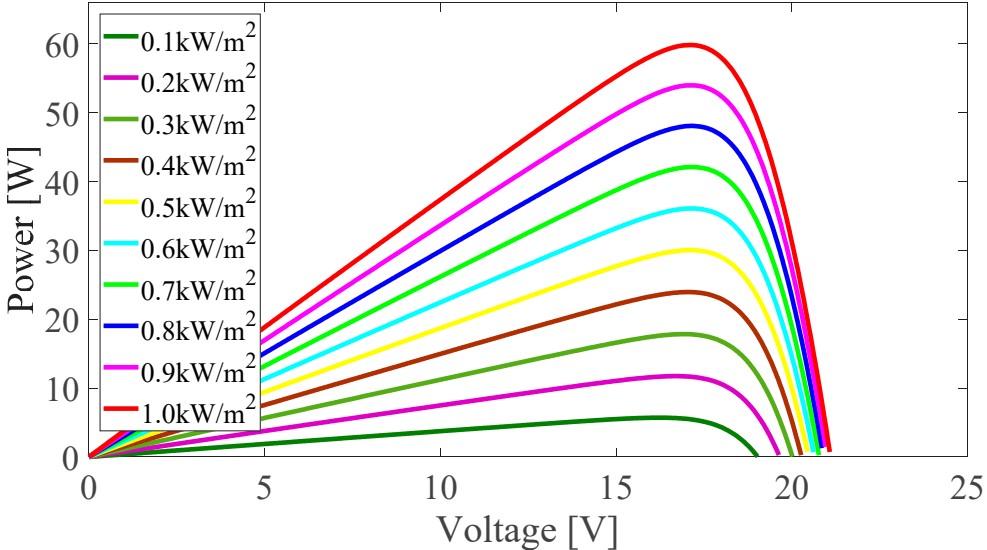

**Figure 8.** Variation of power while the irradiation changes.

The selected of $I_{TH}$ in the Idle Mode is based on the change of MPP current in the MPP region, as shown in Figure 9. At the MPP, the relational gain *KIsc* between MPP current *Impp* and short-circuit current *Isc* is a constant, and $0.78 < KIsc < 0.92$ [56]. In this condition, the *Isc* can be estimated using the listing formula in the left of the MPP.

$$Isc = I - \frac{I - I\_old}{V - V\_old} \cdot V \tag{6}$$

*KIsc* is selected as 0.92 in the simulation.

$I_{TH}$ is expressed as follows:

$$I_{TH} = K_{Isc}\left(I - \frac{I - I\_old}{V - V\_old} \cdot V\right) \tag{7}$$

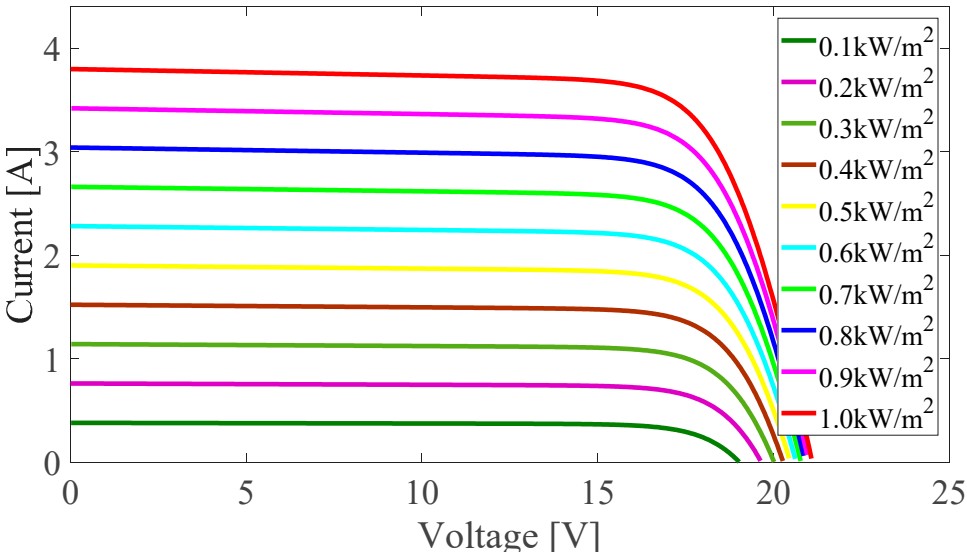

**Figure 9.** Variation of the MPP current.

### 2.3. Simulation Modeling and Power-Loss Analysis

In this part, the simulation modeling and the mathematical method for the power-loss analysis are introduced. To verify the proposed MPPT tactic, a MATLAB/Simulink module is used as a platform for simulation. During the simulation, some parameters are changed to simulate the change in the environmental conditions. The power-loss analysis and calculation are expressed by mathematical equations.

### 2.4. Simulation Modeling

Simulation modeling mainly includes three key aspects: (1) PV module modeling, (2) MPPT controller modeling, and (3) PV system combination. The modeling strategies and parameter settings are presented in the tables and figures.

#### 2.4.1. PV Module Modeling

The modeling of the PV array module is based on the template BP MSX-60W1 from Simulink Library. The symbol and connection are displayed in Figure 10, and the parameters are explained in Table 2.

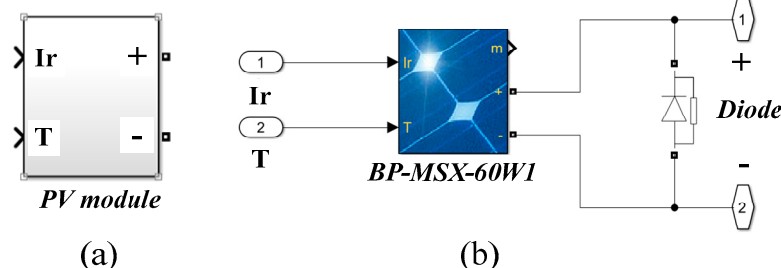

**Figure 10.** PV array module modeling (**a**) symbol and (**b**) internal connection.

In Figure 10, Ir is the input for the illumination intendancy, T is the input for the temperature, "+" is the positive electrode of the output voltage, and "−" is the negative electrode of the output voltage. The diode in Figure 10b protects the PV panel.

**Table 2.** Installed characteristic parameters of the PV array module

| Parameter | $N$ [cell] | $P_{MPP}$ [W] | $V_{OC}$ [V] | $V_{MPP}$ [V] | $I_{SC}$ [A] | $I_{MPP}$ [A] | $C_{VOC}$ [%/°C] | $C_{ISC}$ [%/°C] |
|-----------|------------|---------------|--------------|---------------|--------------|---------------|------------------|------------------|
| Value | 36 | 59.85 | 21.1 | 17.1 | 3.8 | 3.5 | −0.379 | 0.065 |

In Table 2, $N$ is the number of cells per module, $P_{MPP}$ is the maximum power, $V_{OC}$ is the open-circuit voltage, $V_{MPP}$ is the voltage at the MPP, $I_{SC}$ is the short-circuit current, $I_{MPP}$ is the current at the MPP, $C_{VOC}$ is the temperature coefficient of $V_{OC}$, and $C_{ISC}$ is the temperature coefficient of $I_{SC}$.

### 2.4.2. MPPT Controller Module Modeling

To compare the conventional and proposed MPPT tactics, two MPPT controller modules were built. Figure 11 shows the connections of the proposed MPPT control. The codes in the m-functions are based on the flowcharts shown in Figure 7.

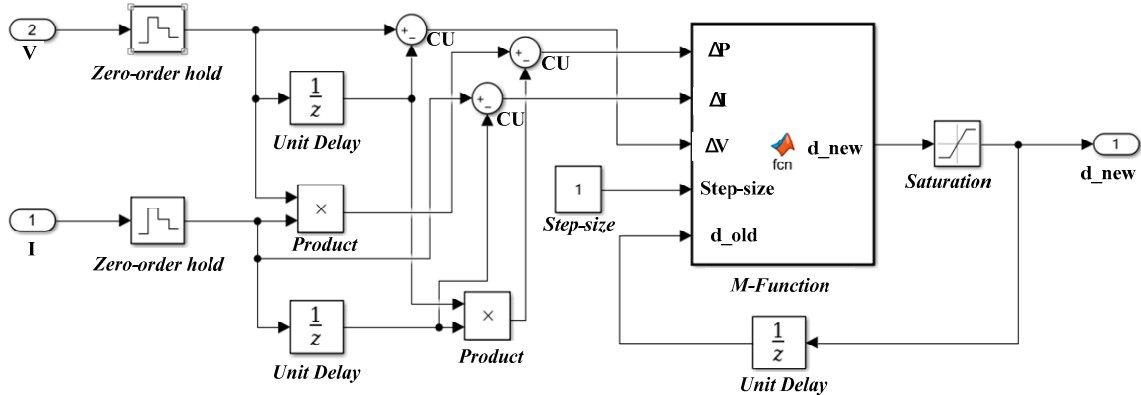

**Figure 11.** Internal connections of the proposed MPPT tactic.

In Figure 11, V is the input data of voltage, I is the input data of current, "zero-order hold" is for updating and holding data at every sample time, "unit delay" is for memorizing data, "product" is a multiplier, "CU" is the unit for data minus, "step size" is for setting the initial value of the step size, "M-function" is a function builder that employs the m-language (five input ports: three for the change in power, current, and voltage; one for initial step size setting; and one for the old duty cycle), "saturation" is for limiting the upper and lower values of a signal, and "d_new" is the updating duty cycle of the PWM wave.

### 2.4.3. PV System Combination

According to the basic structure of the PV system displayed in Figure 1, the proposed P&O-based PV system simulation platform is shown in Figure 12.

The PV module is introduced in the PV array modeling section. The proposed MPPT is explained in the Methods section, and the initial connection is shown in Figure 12. The power converter and load include a boost converter and a 30-Ω resistor (as the electrical appliance); Ir is an input port for the illumination, T is an input port for the temperature, the I sensor is for measuring the photocurrent, the V sensor is for measuring the photovoltage, and C is a filter capacitor (47 μF). The repeating sequence and relational operator work together and generate the control signal (PWM wave) for the power converter. The connections are based on the initial characteristics of each component.

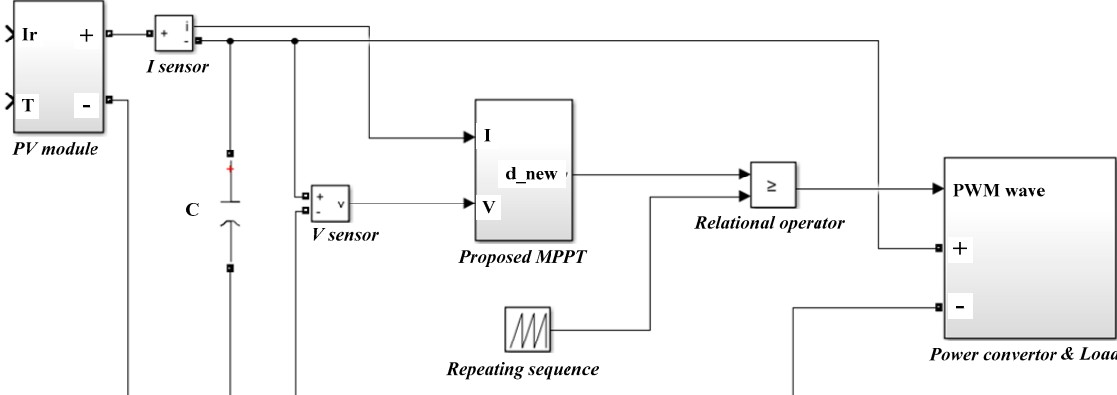

**Figure 12.** Simulation platform of the proposed MPPT tactic. PWM—pulse-width modulation.

### 2.5. Power-Loss Analysis and Calculation

The power-loss analysis is an important aspect for defining the tracking efficiency of the MPPT. The artificial oscillation around the MPP of the proposed MPPT strategy is diminished to close to zero and can even be removed in an ideal environment; moreover, the tracking step size can be adapted automatically. A typical operation issue of the conventional P&O MPPT strategy is displayed in Figure 13, and the power-loss analysis and calculation are based on this figure.

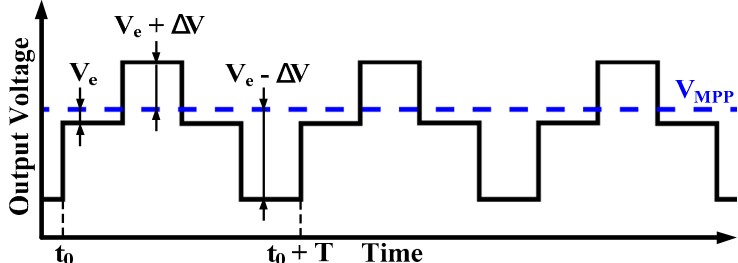

**Figure 13.** Typical operation issue analysis of ordinary P&O.

The relationship between the power loss ($P_L$) and the theoretical power ($P_{MPP}$) is expressed as follows [57]:

$$\frac{P_L}{P_{MPP}} \approx \left(\frac{(\Delta V_{PV})_{RMS}}{V_{MPP}}\right)^2 \left(1 + \frac{V_{cell}}{2nkT/q}\right) \tag{8}$$

Here, $V_{MPP}$ is the theoretical output voltage when the PV panel operates in the MPP, $(\Delta V_{PV})_{RMS}$ is the root-mean-square [58] value of the voltage perturbation, and $V_{cell}$ is the output voltage of every single cell when the PV panel operates in the MPP (mostly around 0.5 V).

According to Figure 13, in one oscillation cycle ($T$), the function of the change in the output voltage corresponding to time ($\Delta V_{PV}(t)$) is expressed as follows.

$$\Delta V_{PV}(t) = \begin{cases} V_e & (t_0 < t < t_0 + T/4) \\ V_e + V_{step-size} & (t_0 + T/4 < t < t_0 + T/2) \\ V_e & (t_0 + T/2 < t < t_0 + 3T/4) \\ V_e - V_{step-size} & (t_0 + 3T/4 < t < t_0 + T) \end{cases} \tag{9}$$

Here, $V_e$ is the minimum difference between $V_{MPP}$ and the output voltage ($V_{PV}$) set by the MPPT controller, and $V_{step\ size}$ is due to the step size and is displayed as $\Delta V$ in Figure 12. The relationship between $V_e$ and $V_{step\ size}$ can be observed as follows.

$$V_e = \beta V_{step-size} \tag{10}$$

Here, $\beta$ is a constant. Replacing $V_e$ with Equation (9), Equation (10) can be expressed as follows.

$$\Delta V_{PV}(t) = \begin{cases} \beta V_{step-size} & (t_0 < t < t_0 + T/4) \\ (\beta+1)V_{step-size} & (t_0 + T/4 < t < t_0 + T/2) \\ \beta V_{step-size} & (t_0 + T/2 < t < t_0 + 3T/4) \\ (\beta-1)V_{step-size} & (t_0 + 3T/4 < t < t_0 + T) \end{cases} \tag{11}$$

Therefore, the root-mean-square (RMS) value of the voltage perturbation $((\Delta V_{PV})_{RMS})$ can be calculated as follows.

$$\begin{aligned} (\Delta V_{PV})_{RMS} &= \sqrt{\frac{1}{T}\int_0^T \Delta V_t^2 dt} \\ &= V_{step-size}\sqrt{\frac{1}{4}\left(\beta^2 + (\beta+1)^2 + \beta^2 + (\beta-1)^2\right)} \\ &= V_{step-size}\sqrt{\frac{1}{2} + \beta^2} \end{aligned} \tag{12}$$

By combining Equations (8) and (12), the steady-state power loss can be expressed as follows.

$$\frac{P_L}{P_{MPP}} \approx \left(\frac{1}{2} + \beta^2\right)\left(\frac{V_{step-size}}{V_{MPP}}\right)^2\left(1 + \frac{V_{cell}}{2nkT/q}\right) \tag{13}$$

For the proposed MPPT tactic, the output voltage does not give rise to any artificial oscillation, but differences still exist between $V_{PV}$ and $V_{MPP}$. The difference between $V_{PV}$ and $V_{MPP}$ in this condition can be expressed as follows.

$$(\Delta V_{PV})_{RMS} = \beta V_{step-size} \tag{14}$$

By substituting Equation (14) into Equation (8), the power loss of the proposed MPPT tactic can be expressed as follows.

$$\left(\frac{P_L}{P_{MPP}}\right)_{proposed} \approx \beta^2\left(\frac{V_{step-size}}{V_{MPP}}\right)^2\left(1 + \frac{V_{cell}}{2nkT/q}\right) \tag{15}$$

## 3. Results and Discussion

The results are categorized into two parts: (1) the simulation results are displayed, analyzed, and compared with the conventional P&O MPPT tactic to show the improvement; and (2) the power loss is calculated via the statistical method expressed in the Methods and power-loss analysis and calculation parts of this report.

### 3.1. Simulation Results

To verify the advanced performances of the proposed MPPT controlling tactic, the simulations follow the single-variable principle, and the comparisons are performed under the same parameter settings (excluding the MPPT controller module). The settings of the PV array module and the other basic simulation parameters are shown in Table 3.

**Table 3.** Global simulation parameters.

| Parameter | Temperature [°C] | Step Size [%] | $E_{TH}$ [W] | $I_{TH}$ [A] | $k$ |
|-----------|------------------|---------------|--------------|--------------|-----|
| Value | 25 | 1 | 0.03 | Equation (7) | 0.5 |

The simulation results are divided into three parts: (1) tracking speed comparison under steady environment conditions, (2) reliability under variable environment operation, and (3) improvement of the steady state operation.

### 3.1.1. Tracking Speed Comparison

The simulation verifies the increasing tracking speed of the proposed MPPT tactic. The other simulation parameter is set to the ideal value (illumination = 1000 W/m$^2$) to eliminate the effects of environmental conditions.

Figure 14 shows the power–time curves of the proposed MPPT tactic (red line) and the conventional P&O algorithm (green line). According to Figure 14, the time needed for the conventional P&O MPPT tactic is 1.061 s, and that for the proposed MPPT tactic is 0.272 s. The decrease in the tracking time verified the increase in the tracking speed; compared with the conventional P&O tactic, the proposed tactic can reduce the tracking speed by approximately 74.5%.

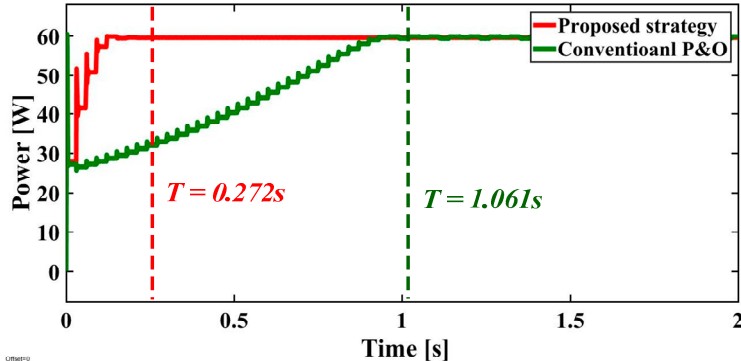

**Figure 14.** Power–time curves of the conventional and proposed MPPT tactics.

### 3.1.2. Reliability under Variable Environmental Conditions

In actual operation, the change in temperature is not sharp, and the main influencing factor is the change in illumination because of partial shading; therefore, the reliability of the proposed tactic is defined via simulation in the environment with a variable change in illumination, as shown in Figure 15.

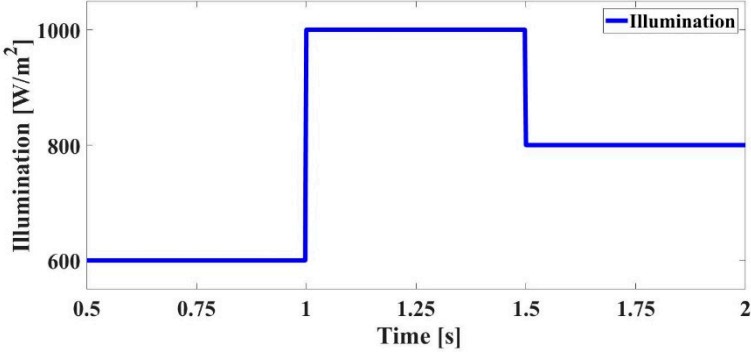

**Figure 15.** Change curve of the illumination.

In this situation, the simulation results, including the current, voltage, and power curves, are shown in Figures 16–18, respectively. In these figures, short explanations of the existing phenomenon are presented. Each figure includes two parts—one is from the conventional P&O tactic, and the other is from the proposed MPPT tactic. Each sub-figure has a standard line of the theoretical output parameters in MPP operation for verifying the tracking accuracy.

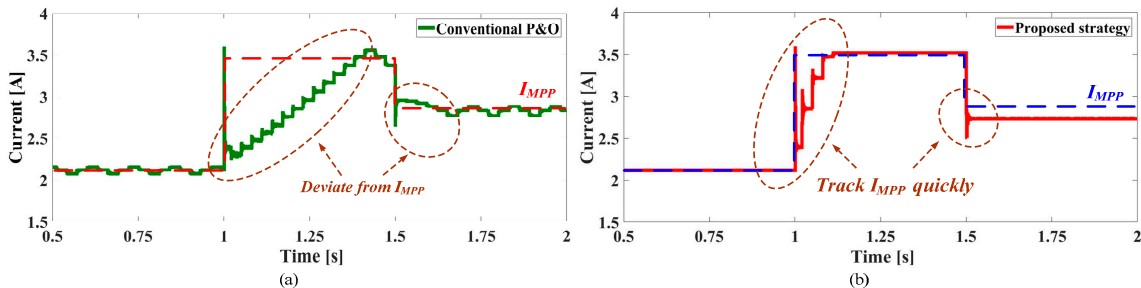

**Figure 16.** Current curves: (**a**) conventional P&O and (**b**) the proposed MPPT tactic.

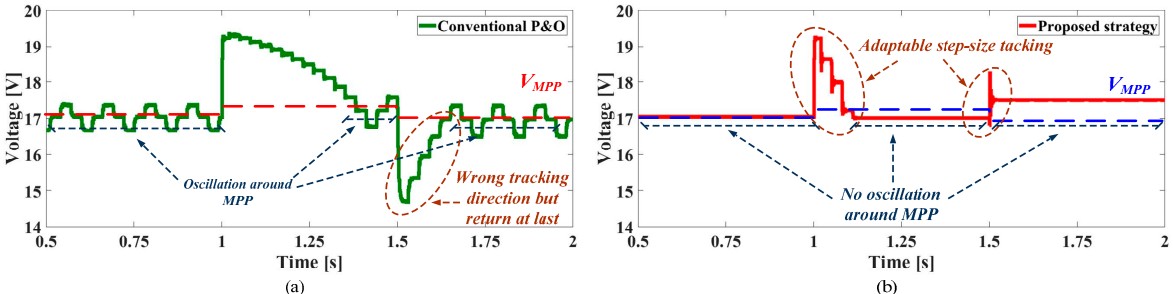

**Figure 17.** Voltage curves: (**a**) conventional P&O and (**b**) the proposed MPPT tactic.

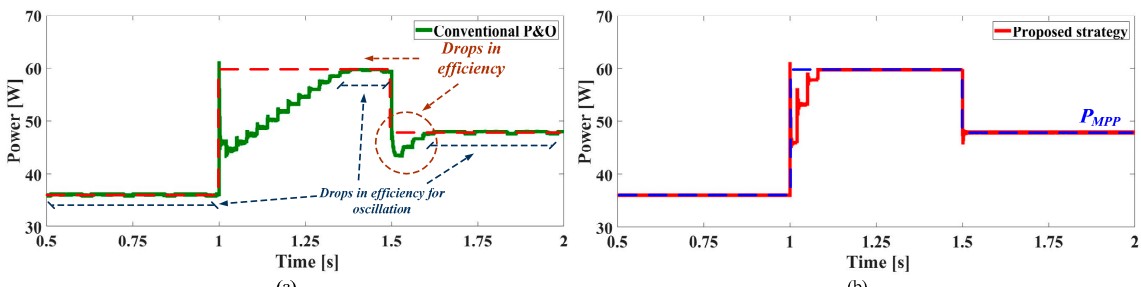

**Figure 18.** Power curves: (**a**) conventional P&O and (**b**) the proposed MPPT tactic.

Figure 16 shows the curves of the PV array output current ($I_{PV}$). Figure 16a shows the measured current for the conventional P&O tactic. The results for the tracking during the increase in the illumination show a lack of speed and departures from the standard line ($I_{MPP}$). As shown in Figure 16b, the proposed tactic does not cause departures and has a high tracking accuracy with the standard.

Figure 17 shows the curves of the PV array output voltage ($V_{PV}$). The conventional P&O tactic can track the MPP, but a loss of efficiency exists in the illumination increasing procedure. This method tracks in the wrong direction owing to the falling illumination and returns to the right direction when this phenomenon stops. During the increase or decrease in the illumination, the proposed strategy operates in the self-adapted step size model and maintains a limited departure from the theoretical output voltage.

Figure 18 shows the curves of the PV array output power ($P_{PV}$). According to the tracking mistakes and errors, the conventional tactic results in efficiency drops, as shown in Figure 18a. At the same time, the oscillation around the MPP causes efficiency drops. As shown in Figure 18b, the output–power curve of the proposed tactic almost coincides with the theoretical output of that of a verification of the tracking accuracy and efficiency.

### 3.1.3. Steady-State Operation Comparison

Figure 19 shows the output–voltage state under steady operation. Compared with the theoretical output value ($V_{MPP}$), there exist conventional P&O strategy oscillations around the MPP in the steady

state; the proposed tactic experiences deviations from the theoretical output, but can maintain operation without oscillation.

### 3.2. Power-Loss Analysis Results

According to Figure 19 and the parameters of the PV module shown in Table 2, the power loss can be calculated using the equation expressed in the Methods section of this report.

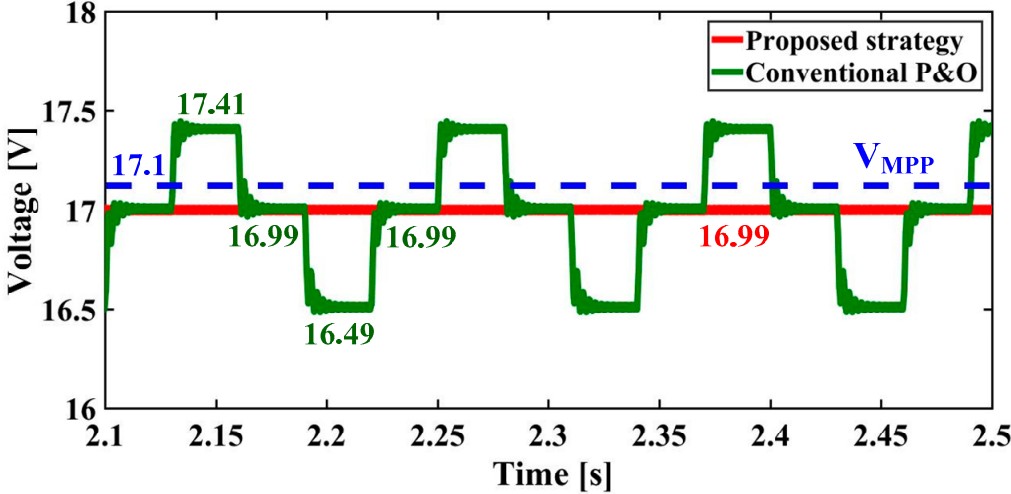

**Figure 19.** Comparison of the voltage in the steady state.

According to Equations (8) and (11), the steady-state power loss of the conventional P&O tactic is calculated as follows.

$$\left( \frac{P_L}{P_{MPP}} \right)_{conventianl} \approx 0.69\% \tag{16}$$

According to Equations (8) and (13), the steady-state power loss of the conventional P&O tactic is given as follows.

$$\left( \frac{P_L}{P_{MPP}} \right)_{proposed} \approx 0.23\% \tag{17}$$

The efficiency of the power-loss reduction is calculated as follows.

$$\eta_{efficiency} = \left( 1 - \frac{0.23\%}{0.69\%} \right) \times 100\% \approx 66.7\% \tag{18}$$

Via the proposed strategy, the power loss in steady-state operation drops to 0.23%; compared with the conventional P&O algorithm, the percentage reduction in the power loss around the MPP is 66.7%.

Assuming the PV element is in the standard test condition (STC) (1000 W/m$^2$, 25 °C, AM1.5), the simulation in steady-state operation for the conventional P&O MPPT and proposed control is as shown in Figure 20.

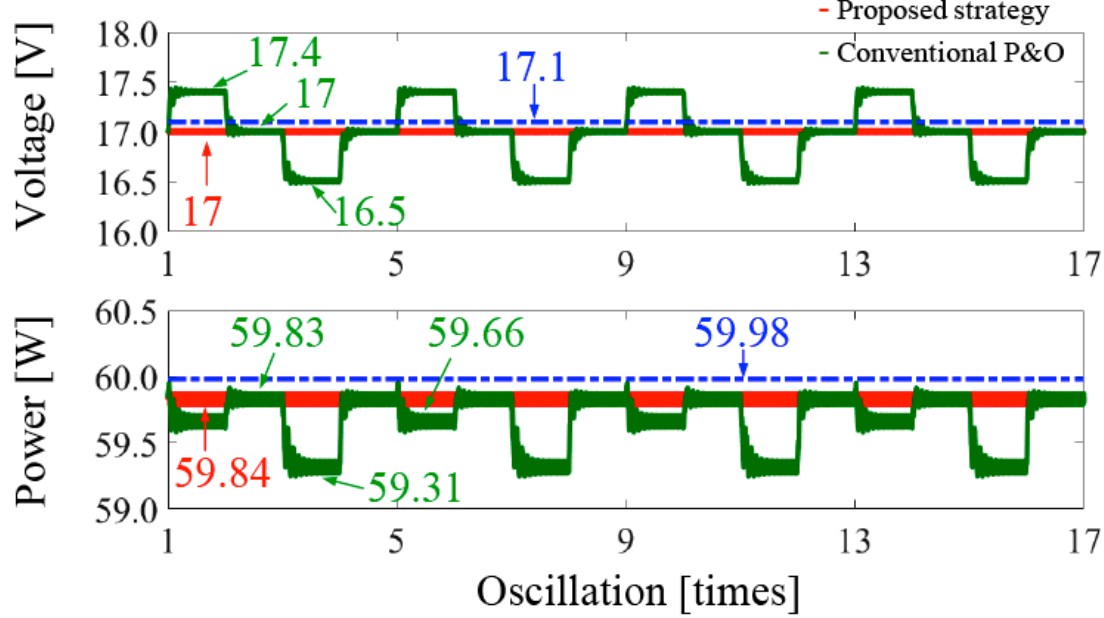

**Figure 20.** Comparison of the voltage and power in the steady state.

The percentage of power absorbed in every oscillation period for the conventional P&O algorithm can be calculated as follows.

$$\frac{P_{t1} + P_{t2} + P_{t3} + P_{t4}}{P_{theo}} = \frac{59.66W + 59.83W + 59.31W + 59.83W}{59.98W \times 4} \times 100\% = 99.46\% \tag{19}$$

where $P_{ti}$ is the $i^{th}$ oscillation in a period and $P_{theo}$ is the theoretical output power during the period.

The power absorbed in every oscillation period for the proposed control scheme can be calculated as follows.

$$\frac{P_{t1} + P_{t2} + P_{t3} + P_{t4}}{P_{theo}} = \frac{59.84 \times 4}{59.98W \times 4} \times 100\% = 99.77\% \tag{20}$$

The power loss in the P&O algorithm is expressed as 0.54% and 0.23% for the proposed control. The simulation result is close to the calculation in the submitted manuscript. The energy saving for every oscillation period in STC is expressed as follows.

$$\sum_{i=1}^{n=4} P_{proposed,ti} - \sum_{i=1}^{n=4} P_{po,ti} = 0.73W \tag{21}$$

The average energy saving in every oscillation is 0.1825W.

A comparison with the conventional P&O algorithm and the theoretical value reveals that the simulation results are well-matched. As the efficiency improves, as shown in Equation (21), the proposed self-adaptable step size MPPT tactic can uncommonly reduce the power loss during the steady-state operation. According to the response speed and tracking accuracy shown in the simulation results at Figure 16 to Figure 18, this proposed tactic can also reduce the power loss during the tracking procedure. Furthermore, the ungraded installations of this proposed tactic are software-based, which means that every PV system with a processor-based MPPT controller can upgrade without any hardware cost.

## 4. Conclusions

This research presents an advanced P&O-based self-adaptable step size MPPT tactic. Compared with the conventional P&O algorithm, this advanced MPPT strategy can reduce the power loss by 0.1825W per oscillation at steady state during the MPP operation; at the same time, the response speed

is lower than 0.3 s, and this strategy has a high stability when facing the slope changing illumination condition. These improvement results are given as follows: (1) the activation of idle operating with the achievement of an allowable tracking error; (2) multiple step size selection; (3) avoidance of natural oscillation; and (4) system operation state determination. The overall performance development, including the steady state and changing illumination operation, verified the benefits of the proposed strategy. These results will contribute to the development of PV installation because the proposed version has higher energy efficiency and reduces the tracking speed and power loss compared with conventional algorithms. In addition to the findings of this study, only numerical calculations show limitations to prove the results. Accordingly, in a future study, an experimental test will be carried out for evaluating the proposed control.

**Author Contributions:** conceptualization, Y.Z., and H.W.; methodology, Y.Z. and M.K.K.; validation, Y.Z.; formal analysis, Y.Z., and H.W.; investigation, Y.Z., and H.W.; resources, Y.Z., and H.W.; Software, Y.Z.; writing original draft preparation, Y.Z. and M.K.K.; writing—review and editing, Y.Z. and M.K.K.; supervision, M.K.K.

**Funding:** This research received no external funding.

**Conflicts of Interest:** The authors declare no conflict of interest.

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
