# Peer review of "Simulation and Analysis of Perturbation and Observation-Based Self-Adaptable Step Size Maximum Power Point Tracking Strategy with Low Power Loss for Photovoltaics"

_energies, doi:10.3390/en12010092_

Round 1
Reviewer 1 Report
The study entitled Perturbation and Observation-based Self-adaptable Step-size Maximum Power Point Tracking Strategy with Low Power Loss for Building-integrated Photovoltaics, presents a self-adaptable step-size P&O-based MPPT algorithm with infinitesimal perturbations, aimed to reduce the power loss in steady-state operation and improve the response speed of MPPT for BIPV.
My general comment are as follows:
Kindly note the hyphenation of the words in the abstract section
Please avoid presenting well established fundamentals (eg section 1.2)
The study presents an analytical model with no validation. Kindly refer to experimental studies of similar content, and try to justify your results.
Please better elaborate the boundary conditions chosen for your analysis.
Reviewer 2 Report
Several words are split without any reason, e.g. maxim-ize, con-ventional, op-eration. Correct.
The authors do not put forward any strong argument to justify the reference to BIPV. They simply discuss another MPPT algorithm which is relevant to any PV module. Either change the title to a more general one or present solid arguments to justify the title.
Reference to “global climate crisis” to introduce an MPPT algorithm is an exaggeration.
Quote references in the text and not at the side of the equations. E.g. page 3/19 Lines 75, 79, 80
The calculation of the percentage losses is quite theoretical and does not reflect actual operating conditions. An approach that would yield reliable results would be to run (1) a simulation over a year and (2) use actual data to determine the losses between the two algorithms.
Quoting a percentage reduction of 66.7% may be misleading with respect to its actual meaning.
The calculations in Tables 4 and 5 are too naive. The authors MUST validate the simulations through experimental data taken over different weather conditions over a year to determine the average annual energy gain, before accepting this work for publication.
Reviewer 3 Report
Reviewer has checked paper and topic of the paper is interesting and suitable for the journal. However paper needs to be improved in several aspect based on the bellow provided comments.
Remarks and recommendations:
1) Some data related to the main research outcomes of the study are missing, so please add sentence,
2) Introduction section needs to be reorganized, i.e. please provide introduction section with clearly defined the main objective of the paper and with quality obtained review (for instance it would be useful also to add a few references related to the general application of PVs such as; Hybrid energy scenarios for residential applications based on the heat pump split-air conditioning units for operation in the Mediterranean climate conditions, Energy and Buildings 140)
3) Section 1.2. and text further should be provided as the separate section named for example (definition of the problem)
4) Section 2: Please clearly stress out what is a novelty of your approach compared with existing MPP concepts?
5) How simulated data correspondes with reality?
6) Conclusion section should contain sentence or two related to the future research work.
Round 2
Reviewer 1 Report
The authors have adequately addressed the review comments.
Author Response
Point 1: The authors have adequately addressed the review comments.
Response 1: We very appreciate your positive comments and hope to contribute our study to the research community to advance science and technology in the domain of Energies.

Reviewer 2 Report
The authors have made certain modifications but still use the term BIPV in their title. However, THIS IS NOT justified anywhere in the text. The paper simply deals with another MPPT algorithm. Nowhere in the paper appears any special design that would support a BIPV application. So it must be removed.
The authors perform a limited number of simulations (not actual all year long experimental data) to conclude in Table 4 about "absorbed solar energy". What is meant by absorbed, what is transition efficiency? This application to the referred cities lacks any scientific support.
Author Response
Point 1: The authors have made certain modifications but still use the term BIPV in their title. However, THIS IS NOT justified anywhere in the text. The paper simply deals with another MPPT algorithm. Nowhere in the paper appears any special design that would support a BIPV application. So it must be removed.
Response 1: Thanks for your keen comment. The changes are newly made in revision paper.
The title is newly revised and corrected to:
Simulation and Analysis of Perturbation and Observation-based Self-adaptable Step-size Maximum Power Point Tracking Strategy with Low Power Loss for Photovoltaics
Point 2: The authors perform a limited number of simulations (not actual all year long experimental data) to conclude in Table 4 about "absorbed solar energy". What is meant by absorbed, what is transition efficiency? This application to the referred cities lacks any scientific support.
Response 2: Thanks for your keen comment. The limited simulation results to conclude in Table 4 and 5 are removed in the paper. And the concrete results verified with experimental data will be analysed and submitted as a next study.
Again, thank you so much for your keen comments and advice.

Reviewer 3 Report
Paper has been properly revised and now acceptable for publication.
Author Response
Point 1: Paper has been properly revised and now acceptable for publication
Response 1: We very appreciate your positive comments and hope to contribute our study to the research community to advance science and technology in the domain of Energies.

Round 3
Reviewer 2 Report
The paper can be accepted for publication. However, the authors should keep in mind that a research work based on simulations is not as solid as when verified by experimental data.